# Prenatal Breastfeeding Counseling Intervention in Women with Pre-Gestational Diabetes Mellitus—A Randomized Controlled Trial

**DOI:** 10.3390/healthcare12030406

**Published:** 2024-02-04

**Authors:** Tal Schiller, Tali Gassner, Yael Winter Shafran, Hilla Knobler, Ofer Schiller, Alena Kirzhner

**Affiliations:** 1Department of Diabetes, Endocrinology and Metabolism, Kaplan Medical Center, Rehovot 7661041, Israel; 2Faculty of Medicine, Hebrew University of Jerusalem, Jerusalem 9160401, Israel; 3Obstetric and Gynecology Department, Kaplan Medical Center, Faculty of Medicine, Rehovot 7661041, Israel; 4Pediatric Cardiac Intensive Care Unit, Schneider Children’s Medical Center of Israel, Petah Tikva 4941492, Israel; 5Faculty of Medicine, Tel Aviv University, Tel Aviv 6997801, Israel; 6Department of Medicine A, Kaplan Medical Center, Faculty of Medicine, Rehovot 7661041, Israel

**Keywords:** diabetes, breastfeeding, pre-gestational diabetes mellitus, pregnancy, lactation counseling

## Abstract

Background: Data on breastfeeding rates and targeted interventions in women with pre-gestational diabetes mellitus are inconclusive. The aim of the study was to evaluate breastfeeding rates up to one year postpartum and whether targeted counseling towards the end of pregnancy can impact breastfeeding rates and duration. An additional goal was to evaluate whether counseling affected women’s perceptions regarding breastfeeding. Methods: Women with pre-gestational diabetes mellitus were cluster-randomized between 32 and 36 weeks of gestation, either to face-to-face instruction with a certified lactation consultant or to receive written information on breastfeeding. Thirty-eight women without diabetes served as controls and were given written information on breastfeeding. All women filled out a questionnaire regarding intended breastfeeding duration, exclusivity, and perceptions, before intervention and at three, six, and twelve months post-partum. Results: Fifty-two women with pre-gestational diabetes mellitus consented to participate. All completed the questionnaires, 26 in each group. At three, six, and twelve months postpartum, rates of any breastfeeding were around 60%, 50%, and 30%, respectively. Approximately one-third breastfed exclusively in each group at three and six months. No significant difference in breastfeeding rates was noted between face-to-face instruction, written information, and controls. End-of-pregnancy counseling improved confidence in breastfeeding knowledge and confidence in being able to manage blood glucose. Conclusions: Breastfeeding rates in pre-gestational diabetes mellitus were comparable to those of women without diabetes and were unchanged by mode of instruction at the end of pregnancy. However, targeted diabetes-oriented breastfeeding instruction at the end of pregnancy improved knowledge and confidence among women with pre-gestational diabetes mellitus.

## 1. Introduction

The World Health Organization (WHO) recommends exclusive breastfeeding for the first six months of life [1]. The benefits of breastfeeding for mother and infant are well established. Breastfeeding mothers have reduced risks of developing subsequent breast cancer, type 2 diabetes mellitus (T2DM), and obesity [2,3]. Breastfeeding may protect infants from infections and reduce the risk of childhood obesity, type 1 diabetes mellitus (T1DM), and T2DM [4,5,6]. Additionally, breastfeeding is effective for weight loss and preventing obesity by increasing energy consumption, which can help exert a positive effect on glucose metabolism by lowering fasting blood glucose and improving insulin resistance in both human and animal studies [7,8]. Consequently, guidelines highly recommend breastfeeding in women with pre-gestational diabetes mellitus [9,10].

Women with pre-gestational diabetes mellitus often encounter additional obstacles to successful breastfeeding that may interfere with or delay initiation, compared to women without diabetes [11,12,13]. Pre-gestational diabetes mellitus increases the risk of maternal and fetal complications (congenital malformations, small for gestational age, large for gestational age, macrosomia, prematurity, respiratory distress, and jaundice) and labor complications (cesarean delivery, shoulder dystocia) [11,12,13].

Also of importance are the increased risks of neonatal hypoglycemia and the likelihood of admission to intensive neonatal care units, which lead to early mother–child separation, and therefore, potentially hindering breastfeeding [14].

Overall, there are limited and inconsistent data regarding breastfeeding rates and duration among women with pre-gestational diabetes mellitus. Some studies demonstrate breastfeeding rates and duration comparable to non-diabetic women [15,16], while others report that women with pre-gestational diabetes mellitus are more likely to forgo breastfeeding or breastfeed for a shorter duration [17,18]. Data on successful interventions to improve breastfeeding rates among women with pre-gestational diabetes mellitus are lacking.

A recent Israeli national survey by the Israeli Center for Disease Control (Israeli CDC) and the Department of Nutrition in the Ministry of Health (MOH) collected questionnaires regarding breastfeeding rates during 2019–2020. The data do not include specific information on women with pre-gestational diabetes mellitus [19]. We found no other data regarding breastfeeding rates in women with pre-gestational diabetes mellitus in Israel.

Regarding the emotional burden of breastfeeding alongside diabetes, studies have shown that during pregnancy, women with pre-gestational diabetes experience reduced general emotional well-being and higher levels of anxiety, depression, worry, pressure, and ambivalence compared to women without diabetes [20,21]. Data likewise suggest that after birth, mothers with pre-gestational diabetes report lower levels of general well-being and lower vitality, compared to women without diabetes. Furthermore, well-being was particularly negatively influenced if breastfeeding affected diabetes management [22]. Consequently, women with pre-gestational diabetes need extensive support from healthcare providers and relatives to initiate and maintain breastfeeding [23,24,25].

Routine breastfeeding counseling in Israel includes general breastfeeding information and guidance given in a group setting in the maternity ward. Upon request, a woman can receive further targeted personal counseling from a maternity ward nurse or an on-staff breastfeeding counselor. When needed, additional counseling can also be received during the newborn monitoring surveillance. However, there is no structured diabetes-specific counseling. If needed, additional breastfeeding guidance and support resources include either attending free or low-cost peer counseling groups when locally available or an out-of-pocket professional breastfeeding counselor’s visit.

Furthermore, data on perceptions, attitudes, and motivations guiding the decision of women with pre-gestational diabetes mellitus to initiate and maintain breastfeeding are also limited. Given the importance of breastfeeding in this population, this understanding is important to design effective interventions.

Consequently, the aim of the study was to evaluate breastfeeding rates up to one year postpartum and whether targeted counseling towards the end of pregnancy could impact breastfeeding rates and duration. An additional goal was to evaluate whether counseling affected women’s perceptions regarding breastfeeding.

## 2. Materials and Methods

### 2.1. Study Design

This prospective randomized study was conducted at the diabetes clinic of a university-affiliated hospital serving as a referral center for pregnant women with pre-gestational diabetes mellitus (either T1DM or T2DM known prior to pregnancy) from the Central–Southern region of Israel. Patients are treated by a multi-disciplinary team, including nurses, dietitians, an endocrinologist, and a maternal–fetal gynecologist. The study was approved by the local Institutional Review Board, and all women signed informed consent forms.

All consecutive women with pre-gestational diabetes mellitus (age ≥ 18 years) attending the diabetes clinic between 1 November 2019 and 30 May 2021 were offered participation. Of 56 eligible women, 52 agreed to participate and were cluster-randomized in groups of 5 women, at 32–36 weeks gestation, to one of two groups: a face-to-face instruction group (where women received personal guidance from a certified lactation consultant experienced in counseling women with diabetes and were provided with written information on dealing with known obstacles to breastfeeding alongside diabetes) and a written instruction group (where women received only general written information and tips about breastfeeding (Appendix A)).

Thirty-eight women without diabetes giving birth at our center served as controls and were given identical written information and tips about breastfeeding as the written information group. Women recruited as controls were healthy with no major illnesses or chronic medications, as verified by the recruiting obstetrician and gynecology physician (Y.W.S.) who was part of the study team. Minor conditions, which, in our opinion, had no bearing on breastfeeding, were allowed. An attempt was made to recruit women of similar ages.

All women had access to standard maternity ward breastfeeding resources; a general postpartum lecture including general guidance on breastfeeding, and, if requested, a meeting with a breastfeeding counselor. All women answered a structured written questionnaire regarding intended breastfeeding duration and exclusivity before intervention, and at three, six, and twelve months postpartum. All questionnaires also recorded women’s concerns and expectations surrounding breastfeeding, motivation, confidence, support systems, and perceptions of diabetes as an obstacle to successful breastfeeding (for those with pre-gestational diabetes mellitus).

The questionnaires were designed by the diabetes clinic team in collaboration with the certified lactation consultant (T.G.(. Before answering the initial questionnaire, a nurse or physician from the diabetes clinic provided participants with a detailed explanation of its contents. Questionnaires were administered personally before intervention and completed online thereafter.

Women were excluded if (1) a fetal abnormality was diagnosed before birth; (2) they had multiple gestations; (3) they had gestational diabetes mellitus (GDM); (4) informed consent could not be given.

### 2.2. Face-to-Face Instruction Content

All face-to-face instructions were conducted according to a uniform topic list by author T.G., a certified lactation consultant, and lasted 30 to 60 min. Instruction content was based on a comprehensive literature review and T.G.’s experience and included major topics known to influence breastfeeding for women with pre-gestational diabetes mellitus. The purpose of the instruction was twofold: to elucidate practical aspects and strategies to overcome challenges to breastfeeding alongside diabetes, and to review realistic breastfeeding expectations for the early postpartum period, thereby reducing confusion and stress in the early postpartum period and increasing maternal confidence by discussing ways to overcome the reviewed challenges. Specifically, the following subjects were discussed: 1. Setting expectations for the nursing mother and baby dyad, such as frequency of feeding, standard amounts of milk per age, and ways to evaluate whether baby is feeding well; 2. Health-related challenges surrounding breastfeeding in the early postpartum months (awareness and management options), such as engorgement, signs of mastitis, and postpartum depression; 3. Breastfeeding challenges frequently encountered by women with pre-gestational diabetes mellitus (awareness and management options), including early mother–baby separation after birth, and the possibility of delayed lactogenesis; 4. Practical aspects of glucose management following labor (i.e., specific tips on glycemic control surrounding breastfeeding, including awareness and management of hypoglycemia).

### 2.3. Written Information Content

The English translation of the original Hebrew written information given to women is presented as Appendix A. The content briefly conveyed major breastfeeding-related topics, as aforementioned in the face-to-face instruction.

### 2.4. Data Collection

Recorded questionnaire data included sociodemographic characteristics (age, education level (i.e., high school graduate, parity), anthropometric data, newborn data (gestational age, weight at delivery), obstetrics and gynecological history (mode of delivery, maternal and fetal complications), and health-related factors (smoking, alcohol consumption). Information obtained regarding breastfeeding included previous experience, duration, and exclusivity.

### 2.5. Definitions

Exclusive breastfeeding was defined as breastmilk only (chestfeeding or pumped breastmilk) provided until six months postpartum and thereafter as breastmilk alongside complimentary solid foods.

Any breastfeeding was defined as any amount of breastmilk alongside additional infant formula milk.

### 2.6. Statistical Analysis

Categorical and nominal variables were reported as frequency and percentages, and continuous variables were reported as means and standard deviation or as medians and ranges. Continuous variables between study groups were tested for normality by the Shapiro–Wilk test. When an abnormal distribution was found, the Mann–Whitney test was performed to compare two groups, and the Kruskal–Wallis test was performed to compare more than two groups. The *T*-test was used to compare variables with normal distribution. Categorical and nominal variables were analyzed by Pearson’s chi-square (χ^2^) test or Fisher’s exact test. Testing for significance before and after instruction was performed using the McNemar test.

## 3. Results

### 3.1. Study Population Characteristics

As demonstrated in Table 1, of the 52 women with pre-gestational diabetes mellitus, 26 women completed the questionnaires in each group; face-to-face instruction, and written information. A total of 38 women without diabetes served as controls. In the pre-gestational diabetes mellitus group, 62% had T1DM and 38% had T2DM. Groups were comparable regarding age, nulliparity, and smoking rates. The offspring birth weight did not differ between groups; however, women with pre-gestational diabetes mellitus delivered significantly earlier and had significantly higher rates of cesarean delivery compared to women without diabetes (Table 1).

### 3.2. Breastfeeding Rates

Table 2 demonstrates the intention to breastfeed, as well as breastfeeding rates in the delivery room, and throughout the follow-up period. Past breastfeeding experience across the face-to-face, written information, and control groups was 65%, 58%, and 74% respectively, and almost all women in our cohort intended to breastfeed. Approximately 45% of women in the face-to-face instruction and written information groups attempted to breastfeed in the delivery room. At three and six months postpartum, rates of any breastfeeding were about 60% and 50%, respectively. At twelve months, 38% and 27% were still breastfeeding in the face-to-face instruction and written information groups, respectively. No significant differences in breastfeeding rates were noted between face-to-face instruction and written information groups. When divided into T1DM and T2DM, breastfeeding rates were 72% and 55% at three months, 50% at six months, and 28% and 40% at twelve months, respectively, without other significant differences between groups .

### 3.3. Association Analysis for Successful Breastfeeding at Six Months Postpartum

Half of the 52 women with pre-gestational diabetes mellitus discontinued breastfeeding before six months postpartum, and the other half continued to breastfeed beyond six months (Appendix A). Unsurprisingly, the strongest predictors of continued breastfeeding at six months and beyond were previous breastfeeding experience (88% vs. 35%, *p* < 0.001) and attempted breastfeeding in the delivery room (73% vs. 15%, *p* < 0.001). Additionally, women who had a cesarean delivery tended to discontinue any and all breastfeeding within the first six months (27% vs. 62%, *p* = 0.03). Type of diabetes, age, education level, and being nulliparous were not significant breastfeeding predictors.

### 3.4. Did Face-to-Face Instruction Affect Change among Women with Pre-Gestational Diabetes Mellitus?

Women after instruction felt significantly more confident in their ability to manage blood glucose alongside breastfeeding (27% before vs. 54% after instruction, *p* = 0.039). More importantly, significantly more women felt confident in their breastfeeding knowledge (62% before vs. 89% after instruction, *p* = 0.016). Although modestly, most other parameters the women were asked about also improved. Collectively, the instruction had a meaningful and positive impact on perceptions, opinions, self-perceived knowledge, and confidence in one’s ability breastfeed alongside diabetes (Table 3).

It is noteworthy that, after instruction, women felt significantly more confident reaching out to healthcare professionals when needed (62% vs. 89%, *p* = 0.016) rather than just social media or family and friends, suggesting that in-hospital counseling is foundational in establishing a positive connection with medical personnel.

Additionally, the knowledge gained in the instruction remained significantly helpful to the women over time; immediately after instruction, 73% found the instruction helpful, and 81% and 73% found it still helpful at 3 and 6 months postpartum, respectively.

## 4. Discussion

Our prospective randomized study aimed to evaluate breastfeeding rates and duration up to one year postpartum in Israeli women with pre-gestational diabetes mellitus, as well as the impact of prenatal counseling on these factors. Our results demonstrate that most women planned to breastfeed and that rates of any or exclusive breastfeeding were similar at three, six, and twelve months postpartum.

Breastfeeding rates in our study are comparable to those of the 2019–2020 national survey conducted by the Israeli CDC and the Department of Nutrition in the Ministry of Health. That survey encompassed approximately 1600 healthy mothers who answered a structured questionnaire at one year postpartum. The rates of any breastfeeding were 54.8% at 6 months postpartum and 29.3% at 12 months postpartum. Exclusive breastfeeding rates up to six months postpartum were 28.8% and decreased to 15.3% thereafter [19]. Breastfeeding rates in the survey were similar to those in our cohort, with approximately half of the women breastfeeding at six months postpartum, and around one-third at one year postpartum.

The literature comparing breastfeeding rates in women with and without pre-gestational diabetes mellitus is inconsistent. Some studies agree with our results, demonstrating comparable breastfeeding rates. Webster et al. followed 19 T1DM mothers and 18 non-diabetic women in Australia. Breastfeeding rates from hospital discharge to three months postpartum were comparable between the groups (63% vs. 78% at discharge and 47% vs. 33% at three months postpartum) [15]. Similarly, no differences were found in breastfeeding rates among 102 T1DM Danish women compared to the general population, interviewed five days and four months following delivery [16]. Several studies, however, found lower breastfeeding rates among women with pre-gestational diabetes mellitus. A Swedish cohort study compared 108 women with T1DM and 104 controls, matched for parity and gestational age. Diabetic mothers breastfed significantly less at two and six months postpartum compared to non-diabetic mothers [14]. Similarly, in two German studies, breastfeeding rates of offspring to mothers from the BabyDiab cohort were followed for one year postpartum [15,16]. They compared 997 offspring of T1DM mothers to 563 offspring of T1DM fathers or siblings. A total of 77% of offspring to T1DM mothers were breastfed, compared to 86% of the control group (and with a shorter duration in the T1DM mothers’ group) [19,26]. Finkelstein et al. explored the intention to breastfeed and breastfeeding rates at discharge across pre-gestational diabetes mellitus, GDM, and controls [26]. Women with insulin-treated pre-gestational diabetes mellitus had the poorest outcomes regarding breastfeeding rates, both in intention to breastfeed and at hospital discharge. Women with non-insulin-treated and GDM had lower in-hospital breastfeeding rates, and GDM was associated with lower breastfeeding rates at discharge [26].

Overall, most studies showed lower breastfeeding rates among women with pre-gestational diabetes mellitus. Possible explanations for the different results may stem from differences in cohort size, baseline characteristics of the studied populations, differences in breastfeeding support by local health care services, and great variations in each country’s maternal leave duration.

Importantly, considering that cesarean deliveries tend to lower breastfeeding rates and that women with pre-gestational diabetes mellitus had higher rates of cesarean delivery, the fact that breastfeeding rates were comparable to women without diabetes might indicate that counseling towards the end of pregnancy can positively impact breastfeeding rates and duration in women with pre-gestational diabetes mellitus.

Considering the advantages of breastfeeding for both the women with pre-gestational diabetes mellitus and their offspring, strategies to encourage breastfeeding are extremely important. Sadly, we could not demonstrate an advantage in breastfeeding rates following a face-to-face meeting with a certified lactation consultant at the end of pregnancy compared to written information. Several explanations can be considered as to the reason: (1) most of our cohort were women with past breastfeeding experience. Possibly, targeting nulliparous women would have yielded different results; (2) women were consulted once at the end of pregnancy and not in the postpartum period; (3) breastfeeding rates may have been influenced by the COVID-19 pandemic through various mechanisms, such as extensive lockdowns that increased time spent at home, the loss of available income, the perceived immunological benefits of breastfeeding for the newborn, and in-hospital newborn–mother separation policies.

We could not find any prospective interventional studies on women with pre-gestational diabetes mellitus utilizing targeted breastfeeding counseling. In a Canadian study, however, it was noted that women seeking antenatal care from non-medical providers were two to three times more likely to breastfeed at hospital discharge, emphasizing the importance of antenatal support [26]. Two studies concerning women with GDM examined the benefit of counseling [27,28]. Griffin et al. found that among women with GDM, inpatient lactation counseling in the immediate postpartum period was associated with improved breastfeeding rates [27]. Ferrara et al. randomized women with GDM to lifestyle intervention, including breastfeeding to encourage weight loss [28]. Women in the intervention group were referred to a lactation consultant at the end of the pregnancy. The authors found no differences in breastfeeding rates at six weeks and seven months postpartum between the intervention and control groups [28].

Our data suggest that prior breastfeeding experience and breastfeeding in the delivery room were the strongest predictors for any breastfeeding beyond six months postpartum. Vaginal delivery, as compared to cesarean delivery, was also a significant factor for successful long-term breastfeeding. These factors were previously demonstrated as significantly associated with successful breastfeeding, and efforts should be made to encourage immediate breastfeeding in the delivery room [14].

When considering the benefits of a targeted instruction session for women with pre-gestational diabetes mellitus at the end of pregnancy, we found that such instruction by an experienced certified lactation consultant was beneficial in filling knowledge gaps, had a significant impact on perceived breastfeeding knowledge, and improved confidence in the ability to manage diabetes alongside breastfeeding. Women were also made more aware of breastfeeding advantages for both mother and newborn, and their confidence in receiving sought postnatalhelp from a healthcare professional was improved. The previous literature suggests that women felt they would need support from healthcare providers should difficulties emerge but were less confident in the medical staff’s ability to meet their needs [29]. A literature review conducted by Rasmussen et al. found that women with T1DM experienced a postnatal sense of disconnectedness from health professionals and a focus on the medicalization of pregnancy [30] at a critical time, i.e., when they experienced increased levels of anxiety, diabetes-related distress, and guilt. Consequently, their study stressed the importance of forming a trusting relationship with healthcare professionals, with shared decision-making and diabetes management assistance [30]. An important point arising from the data in our study is that the confidence and trust of these women seeking help from healthcare professionals can be improved. Although women answered they would receive sufficient support from the medical staff after birth if needed, the number of women stating they would, in fact, seek professional guidance increased significantly after instruction. Hence, the instructional intervention can lay the foundation for improved communication between women and the healthcare staff. As meeting continuous diabetes management needs, which do not cease after birth, is critical, promoting optimal trust and communication with healthcare staff is likewise critical. Breastfeeding-related concerns raised by these women should be addressed, and further education and support should be given to healthcare providers caring for women with pre-gestational diabetes mellitus so that relevant assistance can be conveyed to the expectant women themselves.

The benefit of our prenatal intervention seems to be in boosting the confidence of women with pre-gestational diabetes mellitus and in lowering their anxiety regarding breastfeeding and the perceived negative effect of diabetes on breastfeeding. The role of healthcare providers in encouraging and supporting women with pre-gestational diabetes mellitus to breastfeed is enormous.

The strength of our study lies in its well-characterized cohort of both T1DM and T2DM women with a year of follow-up. Study limitations include the relatively small number of women, not allowing for sub-analysis, and the fact that the data were collected from questionnaires with the possibility of recall bias. Further, the questionnaire employed was not previously validated. Taken together, the results of the study indicate that further continuous education and support for women with pre-gestational diabetes mellitus and their healthcare teams are warranted.

## 5. Conclusions

Our results suggest that breastfeeding rates in a population of Israeli women with pre-gestational diabetes mellitus are similar to non-diabetic women. A targeted counseling intervention at the end of pregnancy did not further improve those rates, but did positively affected knowledge and confidence surrounding breastfeeding. The need remains for further studies to find the optimal combination of education and supportremains.

## Figures and Tables

**Table 1 healthcare-12-00406-t001:** Baseline characteristics of study participants.

Characteristics	FTF (n = 26) (a)	WI (n = 26) (b)	Without Diabetes (n = 38) (c)	*p*-Value *
Type 1/Type 2 diabetes, n (%)	12/14 (46/54)	20/6 (77/23)		0.046
Maternal age (years), mean ± SD	35 ± 5	32 ± 5	32 ± 6	0.169 avsb ^†^ 0.304 avsc ^††^ 1.000 bvsc ^†††^
High School graduate, n (%)	11 (42)	19 (73)	24 (63)	0.049 avsb 0.165 avsc 0.576 bvsc
Nulliparous, n (%)	5 (19)	6 (23)	5 (13)	0.942 avsb 0.759 avsc 0.486 bvsc
Smoker, n (%)	4 (15)	2 (8)	5 (13)	0.667 avsb 1.000 avsc 0.456 bvsc
Cesarean delivery, n (%)	12 (46)	11 (42)	2 (5)	1.000 avsb <0.001 avsc <0.001 bvsc
Birth weight (grams), mean ± SD	3359 ± 478	3496 ± 506	3356 ± 498	1.000 avsb 1.000 absc 0.628 bvsc
Gestational age at delivery, median (min–max)	38 (34–39)	38 (37–39.2)	39 (37–41.5)	0.608 avsb 0.003 avsc <0.001 bvsc
Jaundice in newborn, n (%)	3 (12)	6 (23)		0.465
Hypoglycemia in newborn	3 (12)	8 (31)		0.173

FTF, face-to-face instruction; WI, written information; * *p*-value is significant at <0.05; ^†^ avsb, a column versus b column; ^††^ avsc, a column vs. c column; ^†††^ bvsc, b column vs. c column.

**Table 2 healthcare-12-00406-t002:** Comparative analysis of breastfeeding characteristics across FTF, WI, and non-diabetic groups at baseline, three, six, and twelve months postpartum.

Characteristics	FTF (n = 26)	WI (n = 26)	Without Diabetes (n = 38)	*p*-Value *
Breastfeeding previous experience, n (%)	17 (65)	15 (58)	28 (74)	0.406
Breastfeeding in the delivery room, n (%)	11 (42)	12 (46)	20 (53)	0.705
Planned breastfeeding at 32–36 weeks, n (%)	25 (96)	25 (96)	33 (87)	0.265
Planned exclusive breastmilk, n (%)	13 (50)	12 (46)	21 (55)	0.767
Breastfeeding at three months post-partum, n (%)	16 (62)	18 (69)	23 (61)	0.758
Three months exclusive breastmilk, n (%)	8 (31)	12 (46)	13 (34)	0.473
Breastfeeding at six months post-partum, n (%)	14 (54)	12 (46)	21 (58) **	0.637
Six months exclusive breastmilk, n (%)	7 (27)	7 (27)	11 (31) **	0.933
Breastfeeding at twelve months postpartum, n (%)	10 (38)	7 (27)	13 (36) **	0.644
Twelve months exclusive breastmilk, n (%)	7 (27)	3 (12)	3 (8) **	0.108

FTF, face-to-face instruction; WI, written information; * *p*-value is significant at <0.05; ** data available for 36 women only.

**Table 3 healthcare-12-00406-t003:** Comparison of baseline perceptions regarding breastfeeding in women with pre-gestational diabetes mellitus before and after face-to-face instruction.

FTF Instruction (n = 26)	Before Instruction	After Instruction	*p*-Value *
I feel confident in my breastfeeding knowledge			
Agree	16 (62)	23 (89)	0.016
Breastfeeding is simple and easy			
Agree	10 (39)	9 (35)	1.000
Concerns surrounding breastfeeding			
I will not produce enough breastmilk	10 (39)	11 (42)	1.000
I will suffer from breast pain	9 (35)	6 (23)	0.250
I will suffer from breastfeeding-induced hypoglycemia	9 (35)	11 (42)	0.500
I will suffer from mastitis	7 (27)	9 (35)	0.625
I have no concerns	6 (23)	7 (27)	1.000
I expect breastfeeding will help me to:			
Cause a good physical feeling	17 (65)	19 (73)	0.625
Recuperate more quickly from childbirth	11 (42)	16 (62)	0.063
Lose excess pregnancy weight	10 (39)	11 (42)	1.000
Better manage my blood glucose levels	7 (27)	14 (54)	0.039
Other (infant-related expectations)	4 (15)	1 (4)	0.375
I am most motivated to breastfeed by expected			
Infant health benefits of breastfeeding	25 (96)	26 (100)	1.000
Infant emotional benefits of breastfeeding	19 (73)	22 (85)	0.375
Maternal emotional benefits of breastfeeding	14 (54)	15 (58)	1.000
Maternal health benefits of breastfeeding	13 (50)	16 (62)	0.250
If I encounter breastfeeding difficulties, I will			
Seek professional guidance	16 (62)	23 (89)	0.016
Turn to family/friends for guidance	9 (35)	11 (42)	0.625
Turn to social media for guidance	6 (23)	7 (27)	1.000
Manage on my own	6 (23)	8 (31)	0.625
Stop breastfeeding	2 (8)	1 (4)	1.000
I will/do receive sufficient support from my family after birth			
Agree	25 (96)	25 (96)	1.000
I will/do receive sufficient support from the medical staff after birth			
Agree	26 (100)	26 (100)	1.000
Diabetes will affect my ability to breastfeed			
Agree	6 (23)	5 (19)	1.000
I feel confident in my ability to optimally handle my diabetes after birth			
Agree	21 (81)	24 (92)	0.250
I feel confident in my ability to handle both my diabetes and take care of my newborn			
Agree	22 (85)	24 (92)	0.500
The prenatal breastfeeding instruction changed my mind regarding			
My intent to breastfeed		14 (54)	
Belief in my ability to breastfeed		11 (42)	
None		7 (27)	
My planned duration of breastfeeding		6 (23)	

Data presented as number (%); * *p*-value is significant at <0.05.

## Data Availability

Data are contained within the article and Appendix A.

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
