# Peer review of "Prenatal Breastfeeding Counseling Intervention in Women with Pre-Gestational Diabetes Mellitus—A Randomized Controlled Trial"

_healthcare, 2024, doi:10.3390/healthcare12030406_

Round 1
Reviewer 1 Report (Previous Reviewer 2)
Comments and Suggestions for Authors
Dear authors, thank you for considering the comments, greetings
Author Response
Dear reviewer,
Thank for your for your comments.
Reviewer 2 Report (New Reviewer)
Comments and Suggestions for Authors
Title should reflect it's a RCT
Introduction
Define pre-GDM
Line 48 - 50 : No reference(s) provided for this claim.
Line 50 - 53: The references cited refer to T1D & T2D. Similarly Line 57 - 62 also refers to T1D. Is this a study among women with T1D, rather than pre-GDM?
Methodology
Study design - Lack of details on how cluster randomization was done. Lack of details on how the women was deemed to be pre-GDM. Who did the recruitment? Who did the randomization? Was the trial registered? Provide study flow chart.
Written information content - Redundant content with supplementary doc
Results
Table 1 - not sure what authors meant with alphabets like avsb
Table 2 - were within & between group changes considered?
Line 282 - The sample size is way too small for a valid multivariate analysis to be done. Extremely wide 95% CI is an evidence for this.
Avoid using words like "diabetic women". Correctly address them as women with diabetes or women with pre-GDM.
Discussion
Clearly differentiate study strengths & limitations, as well as implications to practice.
Author Response
- Title should reflect it's a RCT.
Response: We thank the reviewer for the comment. The group of women with pre gestational diabetes mellitus were indeed cluster randomized to either face to face counseling or written information. The control group of women without diabetes was recruited based on the lack of significant health conditions as stated in the manuscript (Page 3 lines 114-118). Upon further consultation with the statistician, we concur that the study meets most criteria to be considered a randomized controlled trial which include: randomization, a control group, intervention and follow-up. Although the groups were not blinded, this is not a mandatory requirement for the trial to be considered as an RCT. Therefor, the phrase RCT, was reintroduced to the title.
- Define pre-GDM.
Response: A definition for pre-gestational diabetes mellitus (either known as T1DM or T2DM prior to pregnancy) was added to the text (Page 3, Line 101).
- Line 48 - 50: No reference(s) provided for this claim.
Response: We thank the reviewer for the comment. References were added to the text (Page 2, Line 51).
- Line 50 - 53: The references cited refer to T1D & T2D. Similarly Line 57 - 62 also refers to T1D. Is this a study among women with T1D, rather than pre-GDM?
Response: Thank you. Women recruited for the study were diagnosed with pre gestational diabetes mellitus, either type 1 or type 2 as now added and stated in the methods section (Page 3, Line 101).
- Study design - Lack of details on how cluster randomization was done. Lack of details on how the women was deemed to be pre-GDM. Who did the recruitment? Who did the randomization? Was the trial registered? Provide study flow chart.
Response: We thank the reviewer for the comment. Women were cluster randomized in groups of five by the study team, such that five women received face-to-face instruction and then five women received written instruction. This was added to the text (Page 3, Lines 107-108).
Women were diagnosed with pre gestational diabetes if type 1 or type 2 diabetes were already diagnosed prior to pregnancy. TS from the Diabetes and endocrinology department and YWS (OBGYN) oversaw recruiting the women and signing of informed consent forms.
During the IRB submission at our institution and after consulting with the local IRB committee it was decided that registration of the trial was not mandatory and therefore the trial was not registered.
A flow chart is provided and added as supplementary material.
- Written information content - Redundant content with supplementary doc
Response: We agree with the reviewer. Although the information content provided in the manuscript is not identical to the information provided for non-diabetic women, the two are similar and therefore the information for non-diabetic women was removed from the supplementary material.
- Table 1 - not sure what authors meant with alphabets like avsb
Response: We thank the reviewer for the comment. This form of presentation was suggested by a former reviewer for easier reading. Avsb means a column versus b column. An explanation was added to the tables.
- Table 2 - were within & between group changes considered?
Response: We thank the reviewer for the comment. Within the groups none of the women started breastfeeding again after ceasing to breastfeed. Further, there was no cross-over between the groups. As stated in the limitations section we were unable to perform sub-analysis due to the relatively small sample size (Page 12 lines 426-428).
- Line 282 - The sample size is way too small for a valid multivariate analysis to be done. Extremely wide 95% CI is an evidence for this.
Response: After further discussion with the statistician, we concur with the reviewer and the multivariate analysis was removed.
- Avoid using words like "diabetic women". Correctly address them as women with diabetes or women with pre-GDM.
Response: We thank the reviewer for the comment. We replaced the term "diabetic women" with pre gestational diabetes mellitus throughout the manuscript.
- Clearly differentiate study strengths & limitations, as well as implications to practice.
Response: Study strengths and limitations are stated on Page 12 lines 425-429. Further, a sentence was added at the end of the paragraph indicating implications for practice (Lines 429-431).

Round 2
Reviewer 2 Report (New Reviewer)
Comments and Suggestions for Authors
Thank you for addressing earlier comments. I have a few more concerns to be addressed:
1) Breastfeeding tips ideally should be just a supplementary doc. It is not doing much for the flow of the paper.
2) Table 1 - There are 3 study groups, and based on info presented in Statistical analysis, the authors only considered Mann Whitney, which is a comparison between 2 groups. Kruskal Wallis should have been performed, followed by posthoc comparison.
3) Table 2 - it appears to be a categorical association, e.g. chi-square or Fisher's exact was done. However, the Table caption indicates it's a comparison across timelines. The content of the Table does not support the title, and neither does the analysis. If it's across the timeline, it should have been McNemar, and if it's a comparison of categorical variables with study groups, it would be chi sq or Fisher's exact test. Even when it's the later, I have a problem with multiple p values presented as you should be just done one chi square / fisher's exact test per comparison.
4) Table 3 - before vs after analysis was done but again it was not mentioned in the statistical analysis section. I am not sure if the authors have conducted the correct analysis or did not declare it in the methods.
5) Supplementary Table 1 - The authors used the term "predictive factors". For us to claim this, a multivariate analysis controlling for covariates should have been done. This is not possible with the small sample size in this study. Authors are advised to use the correct term to describe the association analysis.
6) Supplementary Figure 1 is not clear and needs more info.
Author Response
Please find attached the response to the reviewer’s comments

This manuscript is a resubmission of an earlier submission. The following is a list of the peer review reports and author responses from that submission.
Round 1
Reviewer 1 Report
Comments and Suggestions for Authors
Firstly, the idea of the paper is good even if it is a well-known subject. Here are some points:
- It's crucial for women with pregestational diabetes to work closely with their healthcare providers to manage their condition during pregnancy and postpartum. Breastfeeding, along with proper medical care, can contribute to better outcomes for both the mother and the baby. However, individual circumstances may vary, and healthcare providers can provide personalized advice based on a woman's specific health needs and conditions.
- Some examples that you did not include in the paper: breastfeeding enhances insulin sensitivity, breastfeeding burns extra calories, aiding in postpartum weight loss (maintaining a healthy weight is essential for women with pregestational diabetes, as excess weight can worsen insulin resistance), breast milk is rich in essential nutrients and antibodies that help protect the baby against infections and chronic diseases, breast milk is well-balanced and helps stabilize the baby's blood sugar levels, reducing the risk of hypoglycemia, breastfeeding has been associated with a lower risk of childhood obesity.
- As you stated the number of patients is small
- Can you explain how you excluded gestational diabetes?
- Can you explain the impact of the statistics on a questionnaire on this subject?
- Some of your findings like: „offspring birthweight did not differ between groups” line 151-152 or „PGDM delivered significantly earlier” line 152 and „higher rates of cesarian delivery” line 153 – are normal findings in literature.
- Use of too many acronyms
- Past breastfeeding experience being similar between groups is a bias
- How did you define the control group besides just no being diabetes? Did you take into consideration other pathologies that may interfere with breastfeeding?
- With comparable breastfeeding rates my question is: how did this study help?
- Most of centers provide a good lactation consultant for all women
- Rethink the study design
- Do not compare questionnaires on written information with questionaries on counseling because it can lead to bias
Author Response
Please find attached response to reviewer 1

Reviewer 2 Report
Comments and Suggestions for Authors
Estimados autores: A continuación encontrarán los comentarios que hemos realizado con respecto a la propuesta que presentaron.
There is an inconsistency between the type of study indicated in the title (A randomized controlled trial) and that described in the Materials and methods section (prospective cohort study), which needs to be corrected.
As it is a randomized controlled trial, the registration number in ClinicalTrials.gov or other database of national or international clinical trials registries is not included in the document.
On the other hand, the materials and methods section also states that the questionnaires were designed by the research team, but does not specify whether a pilot study was carried out beforehand to validate the instruments.
A predictive factor is described as a variable that increases the risk of presenting with a disease or outcome.
In this sense, in the proposal, the authors describe as strong predictors some variables that showed a difference in the bivariate analysis, but one would expect an irrigation estimator to be reported (HR or RR).
In Table 1, it is suggested to indicate in the column of the obtained p-value which statistical test was used, since only Chi-square is mentioned and values are reported in terms of mean, median and frequency with the corresponding dispersion measure.
The presentation of the tables is understandable. However, another presentation proposal is sent as a suggestion
|
Table 1. Baseline characteristics of study participants. |
||||
|
Variable |
Face to face instruction(a) (n=26) |
Non-frontal-instruction(b) (n=26) |
Without diabetes(c) (n=38) |
p-value |
|
Maternal age (years), media±SD |
35±5 |
32±5 |
32±6 |
0.169avsb |
|
0.304avsc |
||||
|
1.000 bvsc |
||||
Author Response
Please find attached response to reviewer 2

Reviewer 3 Report
Comments and Suggestions for Authors
The manuscript entitled ”Prenatal breastfeeding counseling intervention in women with pre-gestational diabetes mellitus: A randomized controlled trial” is a well-wrtitten and valuable paper. I have only one remark: Authors should take into consideration that PGDM women having higher rates of cesarean delivery compared to women without diabetes were at risk to have lower breast-feeding rates. The obtained results showing comparable BFR in PGDM and control women might indicate that consueling towards the end of pregnancy can positively impact BFR duration in PGDM women.
Author Response
Please find attached response to reviewer 3

Round 2
Reviewer 1 Report
Comments and Suggestions for Authors
Accept in present form
Reviewer 2 Report
Comments and Suggestions for Authors
Dear authors, when I pointed out the inconsistency in the type of study referred to, it was not only to remove from the title that it was a study (A randomised controlled trial), but to clearly define the type of study, the inconsistency remains even in your response, since it points out that the groups were divided based on the "intervention", a cohort study is defined as an observational study in which the population is not exposed to any intervention and the population is divided based on the presence or absence of the exposure or factor.
this sentence: A predictive factor is described as a variable that increases the risk of a disease or outcome. It is part of observation 5, where it is noted that if it is a cohort study, it would be expected that they would report the value of the risk estimate obtained (RR or OR) and not just the p-value, it was added that a logistic regression model was performed, but not the value of the risk estimate (e.g. crude OR and its 95% CI, they only refer to a p-value), nor was the corrected OR and 95% CI interval in the multivariate model reported, indicating only that it remained significant, these data cannot be omitted. ideally include a table where these values are reported.
It is noteworthy that it is mentioned that the variable education was included in the model and in the results it is reported as not significant, as well as breastfeeding in the past and this variable is not found in the results tables, we assume that it refers to (breastfeeding previous experience), so it is suggested to describe the variables in the same way so as not to cause confusion to the readers, which was also not significant in the bivariate analysis, so it is suggested to specify and justify why these variables were chosen to be included in the multivariate model.
Finally , when an instrument or questionnaire is created de novo by a group of experts (usually an odd number), the first step is validation, which in this case means that no prior validation study has been done, which may have a negative impact on the validity and reliability of the results.